# Evaluation of Text-to-Video Generation Models: A Dynamics Perspective

**Mingxiang Liao**[1][*] **Hannan Lu**[2][*] **Xinyu Zhang**[3,4][*] **Fang Wan**[1][†] **Tianyu Wang**[1]
**Yuzhong Zhao**[1] **Wangmeng Zuo**[2] **Qixiang Ye**[1] **Jingdong Wang**[4]

[1]University of Chinese Academy of Sciences [2]Harbin Institute of Technology
[3]The University of Adelaide [4]Baidu Inc.
{liaomingxiang20, wangtianyu21, zhaoyuzhong20}@mails.ucas.ac.cn
{wanfang, qxye}@ucas.ac.cn, {luhannan, wmzuo}@hit.edu.cn
xinyu.zhang02@adelaide.edu.au, wangjingdong@baidu.com

## Abstract

Comprehensive and constructive evaluation protocols play an important role when developing sophisticated text-to-video (T2V) generation models. Existing evaluation protocols primarily focus on temporal consistency and content continuity, yet largely ignore dynamics of video content. Such dynamics is an essential dimension measuring the visual vividness and the honesty of video content to text prompts. In this study, we propose an effective evaluation protocol, termed DEVIL, which centers on the dynamics dimension to evaluate T2V generation models, as well as improving existing evaluation metrics. In practice, we define a set of dynamics scores corresponding to multiple temporal granularities, and a new benchmark of text prompts under multiple dynamics grades. Upon the text prompt benchmark, we assess the generation capacity of T2V models, characterized by metrics of dynamics ranges and T2V alignment. Moreover, we analyze the relevance of existing metrics to dynamics metrics, improving them from the perspective of dynamics. Experiments show that DEVIL evaluation metrics enjoy up to about 90% consistency with human ratings, demonstrating the potential to advance T2V generation models. Project page: t2veval.github.io/DEVIL/.

## 1 Introduction

With the rapid progress of video generation technology, the demand of comprehensively evaluating model performance continues to grow. Recent benchmarks [27, 24] have included various metrics, *e.g.*, generation quality, video-text alignment degree, and video content continuity, to evaluate T2V generation models. Despite of the great efforts made, an essential characteristic of video, *i.e.*, *dynamics*, remains ignored.

Dynamics is a crucial dimension when evaluating video generation models for the following two reasons: ($i$) In practical applications, it is expected that generated video content is honest to text prompts, *e.g.*, dramatic text prompts result in videos of high dynamics. ($ii$) In real-world scenarios, dynamics are negatively relevant to commonly used evaluation metrics, as observed by recent benchmark studies [24, 27]. This allows T2V models to 'cheat' by generating low-dynamic video content in many cases to achieve high scores upon these metrics.

In this study, we introduce DEVIL, a comprehensive evaluation protocol, which assesses T2V generation models from a perspective of dynamics. DEVIL treats *dynamics* as a primary dimension for T2V model evaluation, as well as enhancing the completeness of existing metrics. To fulfill

---

[*] Equal contribution. [†] Corresponding Author.

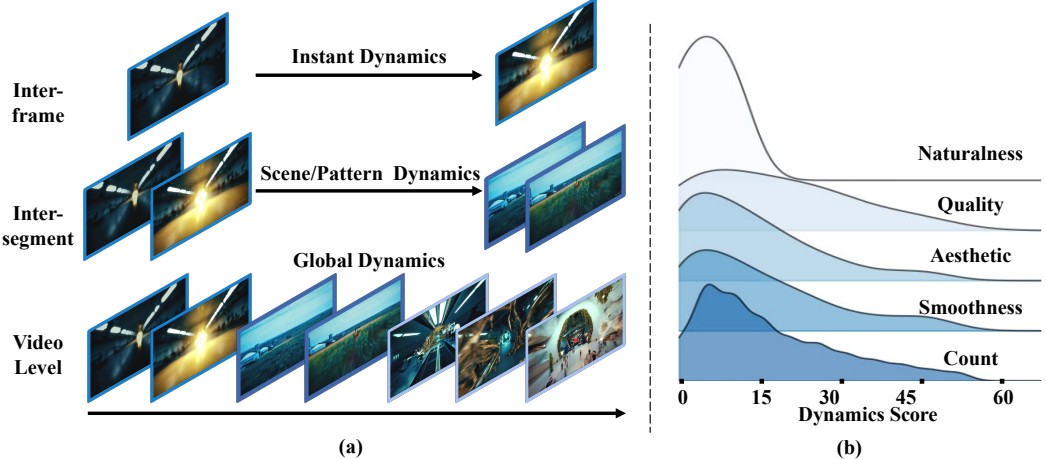

Figure 1: Evaluation of video dynamics. (a) Illustration of dynamics at multiple temporal granularities. (b) Video quality distribution $w.r.t.$ dynamic scores. (Best viewed in color)

this purpose, we first establish a benchmark incorporating text prompts under multiple dynamics grades. The text prompts are collected from commonly used datasets [7, 6, 46, 41] and categorized using a Large Language Model (LLM), GPT-4 [30], with further manual refinement. We then define a set of dynamics scores, which are aggregated into two dynamics metrics to reveal the temporal characteristics of generated videos. In addition, we conduct a user study to synthesize the proposed metrics into an overall dynamics score, facilitating a comprehensive evaluation of dynamics capacity, characterized by dynamics ranges and T2V alignment.

To enhance the completeness of existing evaluation metrics, we introduce a bi-variate analysis strategy. Specifically, we use dynamics as an additional dimension to examine the distribution of existing metrics, such as aesthetics, continuity, and consistency. Through bi-variate analysis, we identify those metrics negatively related to dynamics, and update them by incorporating the dynamics factor. We also introduce a metric to evaluate the naturalness based on a multimodal large language model (MLLM), $e.g.$, Gemini-1.5 Pro [1].

With the proposed DEVIL protocol, we evaluate state-of-the-art T2V models and commonly used benchmarks and find the following problems. ($i$) Existing generation models typically generate slow-motion videos, as most videos in existing benchmarks are of low dynamics. ($ii$)The text prompts in commonly used T2V benchmarks can not reflect the degrees of video dynamics. If such prompts are used to generate videos, they cause poor T2V alignment $w.r.t.$ dynamics. ($iii$) By experiments, we observe that the naturalness of generated video decreases with video dynamics, which implies that the capability of simulating real-world scenarios remains to be elaborated.

The contributions of this study are summarized as follows,

- We propose a novel evaluation protocol, termed DEVIL, which benchmarks T2V generation models by integrating dynamics metrics. Together with existing evaluation metrics, DEVIL builds a more comprehensive evaluation protocol.
- We establish a text prompt benchmark $w.r.t.$ dynamics grades and propose a set of metrics to evaluate video dynamics across temporal granularities, facilitating the assessment of dynamics range and T2V alignment.
- Extensive evaluation of existing T2V generation models allows us to thoroughly analyze the capabilities of T2V models through the proposed protocol and benchmarks. The results would inspire sophisticated T2V generation models.

## 2 Related Work

### 2.1 Text-to-Video Generation Model

As a recent breakthrough in artificial intelligence, diffusion models have pushed video generation technology to a new height. Earlier studies [22, 21] explored the 3D U-Net and cascaded models for

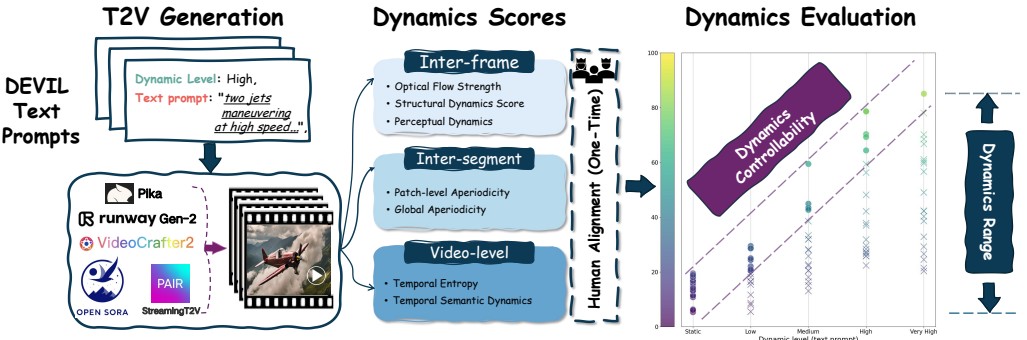

Figure 2: Flowchart to calculate dynamics metrics based on dynamics scores and text prompts.

diffusion within pixel space. Recent solutions [13, 33] employed latent diffusion models to efficiently manage the diffusion process within a compressed latent space. Following these studies, a variety of approaches [40, 10, 26, 43, 16, 42, 47, 29, 25] updated and improved this paradigm. Building on these advancements, subsequent methods further explored generating videos of higher quality and extended duration. The Videocrafter approach [14] pursued high-quality video generation through disentangling spatial and temporal learning and tuning spatial modules using high-quality images. In a similar way, commercial models such as Pika [4] and GEN-2 [2] demonstrated substantial improvements, showcasing videos with exceptional visual clarity. For longer video generation, Gen-L-Video [39] aggregated short clips generated by base T2V models using temporal co-denoising to enhance continuity. Freenoise [31] extended pre-trained T2V models through rescheduling noise for longer-duration video inference. StreamingT2V [19] enhanced long-term content consistency by integrating short-term and long-term memory blocks.

The rapid development of T2V models poses a growing demand for quality evaluation protocols. Unfortunately, existing protocols primarily focus on temporal consistency and content continuity, yet largely ignore temporal dynamics. This hinders the exploitation of video content vividness and the honesty of video content to text prompts.

## 2.2 Evaluation Protocol

Early evaluation protocols [35] primarily relied on class labels to T2V models. For example, they commonly used video clips from the UCF-101 dataset and human-annotated video captions from the MSR-VTT [46] dataset as the evaluation data. For a more specific assessment, FETV [28] assigned fine-grained category labels to prompts and calculated the CLIP-SIM score for each category.

However, conventional quality assessment metrics such as Inception Score (IS) [34], Fréchet Inception Distance (FID) [20], Frechet Video Distance (FVD) [37], and CLIP-SIM typically operate on a single dimension while can not provide a comprehensive evaluation. When addressing the limitation, EvalCrafter [32] expanded both the prompt scale and the number of evaluation metrics so that the text-video alignment degree and the quality of generated videos can be better evaluated. Additionally, VBench [24] proposed a multi-dimensional, multi-category evaluation suite that not only considered the diversity of prompts but also encompassed a variety of assessment metrics.

Despite of the evolution of evaluation metrics, we argue an essential characteristic of video, $i.e.$, dynamics, remains ignored. In this study, we introduce the dynamics dimension to evaluate T2V generation models, as well as enhance the completeness of existing metrics.

## 3 Dynamics Evaluation Protocol

As shown in Fig. 2, we first establish a benchmark incorporating text prompts under multiple dynamics grades. The text prompts are collected from commonly used datasets [7, 6, 46, 41] and categorized to dynamics grades using GPT-4 [30] and human refinement. We then define a set of dynamics

Table 1: Formulations of dynamics scores at different temporal granularities.

| Granularity | Dynamics scores | Formulation |
|---|---|---|
| Inter-frame | Optical Flow Strength | $\mathbf{D}_{ofs} = \frac{1}{N-1} \sum_{i=1}^{N-1} \text{FLOW}(f_i)$ |
| | Structural Dynamics Score | $\mathbf{D}_{sd} = 1 - \frac{1}{N-1} \sum_{i=1}^{N-1} \text{SSIM}(f_i, f_{i+1})$ |
| | Perceptual Dynamics Score | $\mathbf{D}_{pd} = \frac{1}{N-1} \sum_{i=1}^{N-1} \text{PHASHD}(f_i, f_{i+1})$ |
| Inter-segment | Patch-level Aperiodicity | $\mathbf{D}_{pa} = 1 - \frac{1}{HW} \sum_{h,w} \mathbf{ACF}(\{F_{i,h,w}\}_{i=1}^N)$ |
| | Global Aperiodicity | $\mathbf{D}_{ga} = 1 - \frac{1}{\lfloor rN \rfloor} \sum_{i=1}^{\lfloor rN \rfloor} \sum_{j \neq i} \mathbf{SIM}(F_i^r, F_j^r)$ |
| Video | Temporal Entropy | $\mathbf{D}_{te} = \mathbf{H}(f_1, f_2, \cdots, f_N | f_1)$ |
| | Temporal Semantic Diversity | $\mathbf{D}_{tsd} = \frac{1}{N} \sum_{i=1}^{N} \|F_i - \bar{F}\|^2$ |

scores corresponding to multiple temporal granularities, to reveal the video characteristics at multiple temporal levels. We finally conduct a user study to aggregate the proposed dynamics into overall dynamics metrics about ranges of dynamics and T2V alignment.

## 3.1 Text Prompt Benchmark

Building the benchmark includes a coarse categorization step and a post-processing step. In the coarse categorization step, the GPT-4 model is used to categorize approximately 50k text prompts collected from existing benchmarks into five grades. These benchmarks include VidProm [41], WebVid [8], MSR-VTT [46], Didemo [18], etc. The five dynamics grades are summarized as follows:

**Static video**: Video content is nearly stationary. *Example*: A man is laying on the ground.
**Low dynamics**: Video content has slow and slight changes. *Example*: A male fencer adjusts his epee mask and prepares to duel with his sparring partner in slow motion.
**Medium dynamics**: Noticeable activity and changes, but relatively smooth overall. *Example*: Tilt up of shirtless sportsman doing pull-ups on bars during cross-training workout at gym.
**High dynamics**: Fast actions and changes. *Example*: A runner explodes out of the starting blocks, racing down the track.
**Very high dynamics**: Extremely rapid and frequent video content changes. *Example*: A medieval siege with catapults launching, walls breaking, soldiers charging, and arrows raining down.

Figure 3: (a) Distribution of dynamics grades for text prompts from DEVIL, Vbench [24], and Eval-Crafter [27]. (b) Word cloud of the text prompt benchmark of DEVIL.

In the post-processing step, we recruit six human annotators to refine the dynamics grades following same criterion. Finally, we sample 800 text prompts at different dynamics grades for a uniform distribution along the grades.

Fig. 3 shows the statistics of the DEVIL benchmark, which contains approximately 800 text prompts, and each dynamics grade includes 19 object categories and 4 scene categories. Unless otherwise specified, all experiments in this paper are conducted on the DEVIL benchmark.

## 3.2 Dynamics Scores

As illustrated in Fig. 4, video dynamics can be classified into three categories based on their temporal granularities: **(i) Inter-frame dynamics**, which describes the variations between successive frames. The dynamics score at this level reflects rapid and prominent content variations. **(ii) Inter-segment dynamics**, which refers to the changes between video segments which contains $K$ video frames.

Define on a middle-level, this score captures middle-speed transitions and motion patterns. **(iii) Video-level dynamics**, which encompasses the overall content diversity and the frequency of changes throughout the video.

**(i) Inter-frame Dynamics Score.** This is further categorized to *optical flow strength*, *structural dynamics* and *perceptual dynamics*.

*Optical flow strength.* We first employ RAFT [49] to estimate the optical flow [9, 36] for each video frame. The mean optical flow magnitudes of each frame are averaged to calculate the optical flow strength of this frame. Averaging the optical flow strength values of all video frames, we have the optical flow strength $\mathbf{D}_{ofs}$ of the video, as

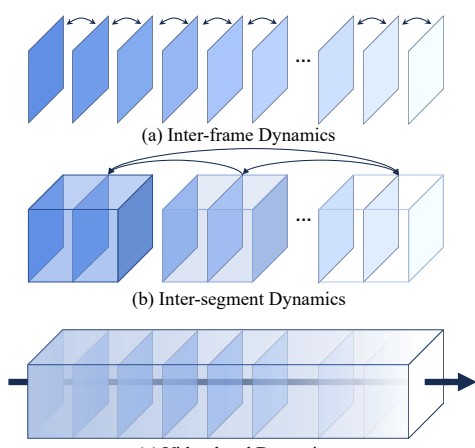

(a) Inter-frame Dynamics

$$\mathbf{D}_{ofs} = \frac{1}{N-1} \sum_{i=1}^{N-1} \text{FLOW}(f_i), \qquad (1)$$

where FLOW calculate the mean optical flow strength values of frame $f_i$.

(b) Inter-segment Dynamics

(c) Video-level Dynamics

*Structural dynamics score.* While optical flow excels in capturing motion, it is less effective when detecting structural dynamics such as lighting conditions. To capture such information, we calculate the average structural similarity index (SSIM) between consecutive frames from all frame pairs to quantify inter-frame structural variations of the video, as

Figure 4: Video dynamics at different temporal granularities: (a) Inter-frame Dynamics, (b) Inter-segment Dynamics, and (c) Video-level Dynamics.

$$\mathbf{D}_{sd} = 1 - \frac{1}{N-1} \sum_{i=1}^{N-1} \text{SSIM}(f_i, f_{i+1})\}_{i=1}^{N-1}, \qquad (2)$$

*Perceptual dynamics.* The human visual system is sensitive to changes in low-frequency regions of video frames. To reflect this characteristic, we introduce a perceptual dynamics metric that measures the difference between the perceptual hashes [38] of consecutive frames. The perceptual distance $D_{pa}$ is defined as the mean of all frame pairs, as

$$\mathbf{D}_{pd} = \frac{1}{N-1} \sum_{i=1}^{N-1} \text{PHASHD}(f_i, f_{i+1})\}_{i=1}^{N-1}, \qquad (3)$$

where PHASHD denotes the Hamming distance [17] between the perceptual hash of $f_i$ and $f_{i+1}$.

**(ii) Inter-segment Dynamics Score.** This is further categorized into *patch-level aperiodicity* and *global aperiodicity*, which measure the dynamics between video segments.

*Patch-level aperiodicity.* We first calculate inter-segment dynamics at the patch level using the auto-correlation factor [11], to evaluate the scene and temporal pattern dynamics. The auto-correlation factor measures the feature similarity of a time series, revealing periodicity and changing trends of features. Given features at position $(h, w)$ across $N$ frames, $\{F_{i,h,w}\}_{i=1}^{N}$, the auto-correlation factor of the features is defined as

$$\mathbf{ACF}(\{F_{i,h,w}\}_{i=1}^{N}) = \frac{1}{N-K_0} \sum_{k=K_0}^{N} \sum_{i=1}^{k} \frac{1}{k} \mathbf{SIM}(F_{i,h,w}, F_{N-k+i,h,w}), \qquad (4)$$

where **SIM** represents the cosine similarity between two feature vectors. The minimal segment length $K_0$ is empirically set to $\lfloor N/8 \rfloor$, as most generated videos contain more than 8 frames. $H$ and $W$ are the height and width of the feature map, respectively. With auto-correlation factors of all patches, we define the patch-level aperiodicity of the video, as

$$\mathbf{D}_{pa} = 1 - \frac{1}{HW} \sum_{h,w} \mathbf{ACF}(\{F_{i,h,w}\}_{i=1}^{N}). \qquad (5)$$

*Global aperiodicity.* In addition to patch-level dynamics, we employ a global aperiodicity metric to measure the diversity of patterns between video segments. Specifically, we divide the video into segments. Each segment has a length $rN$, where $r$ is a proportion factor, empirically set to 0.25. We use ViCLIP [44] to extract the spatial-temporal features for each segment. The features are denoted as $\{F_i^r\}_{i=1}^{\lfloor rN \rfloor}$. We then calculate the similarity of these features to assess the variation in spatial-temporal patterns across segments, as

$$\mathbf{D}_{ga} = 1 - \frac{1}{\lfloor rN \rfloor} \sum_{i=1}^{\lfloor rN \rfloor} \sum_{j \neq i} \mathbf{SIM}(F_i^r, F_j^r). \tag{6}$$

**(iii) Video-level Dynamics.** The dynamics of a whole video sequence is defined upon the *temporal entropy* and *temporal semantic dynamics*.

*Temporal entropy.* To evaluate the dynamics at the video level, we first measure the temporal information of each video. The temporal information is defined as the conditional entropy of the entire video sequence given the first frame

$$\mathbf{D}_{te} = \mathbf{H}(f_1, f_2, \cdots, f_N | f_1). \tag{7}$$

To estimate the conditional entropy $D_{te}$, we employ the video encoding toolbox FFmpeg [15].

*Temporal Semantic Dynamics.* Beyond low-level dynamics, we further introduce a semantic diversity score to assess high-level dynamics across the whole video. The semantic diversity score $\mathbf{D}_{tsd}$ is computed to reflect semantic-level dynamics and is defined as the variance of DINO [12] features $\{F_i\}_{i=1}^N$ of each frame, as

$$\mathbf{D}_{tsd} = \frac{1}{N} \sum_{i=1}^N \|F_i - \bar{F}\|^2, \tag{8}$$

where $\bar{F} = \frac{1}{N} \sum_{i=1}^N F_i$ denotes the mean feature vector of all frames.

### 3.3 Human Aligned Dynamics Scores

To establish a reliable and robust assessment, we introduce a human alignment module, Fig. 2, to refine the empirically defined dynamcis scores. It utilizes human ratings to provide ground-truth, based on which we fit a linear regression model at each temporal level, as

$$\mathbf{D}_f = \mathbf{Linear}_{\theta_f}(\mathbf{D}_{ofs}, \mathbf{D}_{sd}, \mathbf{D}_{pd}), \tag{9}$$
$$\mathbf{D}_s = \mathbf{Linear}_{\theta_s}(\mathbf{D}_{pa}, \mathbf{D}_{ga}), \tag{10}$$
$$\mathbf{D}_v = \mathbf{Linear}_{\theta_v}(\mathbf{D}_{te}, \mathbf{D}_{tsd}), \tag{11}$$

where $\theta_f, \theta_s, \theta_v$ respectively denote the model parameters of linear regression at each scale. [2] The overall dynamics score of the video is then defined as the average of aligned dynamics scores from all three levels, as

$$\mathbf{D} = \frac{1}{3}(\mathbf{D}_f + \mathbf{D}_s + \mathbf{D}_v). \tag{12}$$

Through this learnable human alignment procedure, the empirically defined dynamics scores are more consistent with human perception, as validated in Sec. 5.1.

### 3.4 Dynamics Metrics

After calculating the aligned dynamics scores of all generated videos at inter-frame, inter-segment, and video levels, we combine these scores together to obtain the following two evaluation metrics.

**(i) Dynamics Range.** The metric evaluates model's capability to generate videos with vivid dynamics. A larger dynamics range implies higher dynamics capability. In detail, we determine the dynamics range $\mathbf{M}_{range}$ by identifying the extremes of the dynamic scores over the benchmark, while excluding the top and bottom 1% scores to mitigate the influence of outliers. This is formulated as

$$\mathbf{M}_{range} = \mathbf{Q}_{0.99} - \mathbf{Q}_{0.01}, \tag{13}$$

---

[2]Appendix B presents the model weights and typical values for various dynamics scores.

where $\mathbf{Q}_{0.99}$ and $\mathbf{Q}_{0.01}$ denote the 99th and 1st percentile values of the dynamics scores for videos generated with the DEVIL benchmark, respectively. This metric reflects a realistic spread of dynamics, excluding atypical extremes.

**(i) Dynamics Controllability.** Let $\mathbf{P}^{(i)}, \mathbf{P}^{(j)} \in [1, 5]$ respectively denote the ground-truth dynamics grades of prompt $i$ and $j$, and $\mathbf{D}^{(i)}, \mathbf{D}^{(j)}$ the predicted dynamics scores by prompt $i$ and $j$. For $\mathbf{P}^{(i)} > \mathbf{P}^{(j)}$, we should have $\mathbf{D}^{(i)} > \mathbf{D}^{(j)}$ so that the dynamics scores of the generated videos are consistent with the dynamics grades of text prompts. Accordingly, we can calculate the dynamics controllability metric by

$$\mathbf{M}_{control} = \frac{1}{|T|} \sum_{i=1}^{|T|} \frac{1}{|T| - T_i} \sum_{j:P_j \neq P_i} \mathbb{I}\big((\mathbf{D}^{(i)} - \mathbf{D}^{(j)})(\mathbf{P}^{(i)} - \mathbf{P}^{(j)})\big), \quad (14)$$

where $|T|$ is the total prompt number and $T_i$ denotes the number of prompts at dynamics grade $\mathbf{P}^{(i)}$.

## 4    Improving Existing Metrics with Dynamics

As observed by our experiments, existing metrics have negative relevance to video dynamics. To identify these metrics, we calculate the correlation, *e.g.*, Pearson and Kendal correlation coefficients, between dynamics scores and existing metrics, Table 11. These metrics include naturalness, motion smoothness, subject consistency, and background consistency. Under these metrics, models might 'cheat' for high-quality scores by generating low-dynamic videos.

We improve the identified metrics by incorporating our proposed human-aligned dynamics score $\mathbf{D}$. In specific, we propose to equally divide the human-aligned dynamics score into $L = 13$ intervals. Within each interval, we calculate the mean metric values. The mean values of the $L$ intervals are further averaged as the improved metrics. Upon the improved metrics, to have a high

Table 2: Correlation between the dynamics metric with the existing metrics including **Nat**uralness (Nat), **V**isual **Q**uality [45] (VQ), **M**otion **S**moothness (MS) [24], **S**ubject **C**onsistency(SC) [24] and **B**ackground **C**onsistency(BC) [24]. 'PC' denotes Pearson's correlation, and 'KC' denotes Kendall's correlation.

| Evaluation Metrics | PC | KC |
|---|---|---|
| Naturalness (Nat) | -51.8 | -44.2 |
| Visual Quality (VQ) | -24.8 | -18.6 |
| Motion Smoothness (MS) | -64.0 | -54.6 |
| Subject Consistency (SC) | -88.9 | -74.9 |
| Background Consistency (BC) | -79.4 | -61.4 |

score, the generated videos should spread all dynamics intervals, which implies a large dynamics range. [3]

**Naturalness**. In addition to the improved metrics, we introduce the *Naturalness* metric, which reflects how much the generated videos are like camera-captured ones. This is done by using the MLLM, *i.e.*, Gemini-1.5 Pro [1], to calculate a naturalness score for each video. The scores are categorized into five grades: "Almost Real" (100 points), "Slightly Unrealistic" (75 points), "Moderately Unrealistic" (50 points), "Noticeably Unrealistic" (25 points), and "Completely Fictitious" (0 points). The overall naturalness is then determined by averaging the scores of all videos. For evaluation, we invited five users to rate the naturalness of the generated videos and then perform a correlation analysis between human ratings and model scores. A high correlation (larger correlation coefficients) indicates the plausibility of the naturalness metric.

## 5    Experiment

### 5.1   Human Alignment Assessment

To evaluate the plausibility of the proposed dynamics metrics and the naturalness metric, we conduct the following human alignment experiments.

**Ground-truth Annotation**. We first generate videos using six state-of-the-art (SOTA) T2V models, including GEN-2 [2], Pika [4], VideoCrafter2 [14], Open-Sora [23], StreamingT2V [19] and FreeNoise-Lavie [31] and DEVIL text prompts. For the generated videos, we collect human evaluated

---

[3]Please refer to Appendix D for the details of the improved metrics.

Table 4: Evaluation of dynamics across text-to-video models at multiple temporal levels. Metrics include inter-frame ($\mathbf{M}^f_{range}$), inter-segment ($\mathbf{M}^s_{range}$), and video-level ($\mathbf{M}^v_{range}$) dynamics range and overall dynamics range ($\mathbf{M}_{range}$) also shown. Dynamics ranges and dynamics controllability ($\mathbf{M}_{control}$) are from 0 to 100, where higher scores indicate better performance.

| T2V-Models | Dynamics Ranges | | | | Dynamics Control. |
|---|---|---|---|---|---|
| | $\mathbf{M}^f_{range}$ | $\mathbf{M}^s_{range}$ | $\mathbf{M}^v_{range}$ | $\mathbf{M}_{range}$ | $\mathbf{M}_{control}$ |
| GEN-2 [2] | 18.8 | 49.7 | 21.4 | 27.8 | 80.8 |
| Pika [4] | 36.2 | 56.3 | 29.1 | 36.8 | 72.2 |
| VideoCrafter2 [14] | 38.9 | 43.2 | 17.0 | 29.4 | 56.6 |
| OpenSora [23] | 65.2 | 80.1 | 36.4 | 55.3 | 61.7 |
| StreamingT2V [19] | 59.8 | 80.0 | 69.9 | 61.4 | 64.0 |
| FreeNoise-Lavie [31] | 67.6 | 71.7 | 66.3 | 63.4 | 58.2 |
| VideoCrafter1 [13] | 62.0 | 60.3 | 28.5 | 44.6 | 63.7 |
| Hotshot-XL [3] | 52.2 | 56.8 | 17.6 | 36.1 | 59.0 |
| Show-1 [48] | 55.5 | 62.4 | 37.0 | 45.0 | 74.2 |
| ModelScope [40] | 72.1 | 78.1 | 40.3 | 56.0 | 63.7 |
| ZeroScope [5] | 24.9 | 46.8 | 19.3 | 28.5 | 66.4 |

dynamics and naturalness as the ground-truth. Six persons are recruited to assess each video's grade of dynamics under three temporal levels (Frame, Segment, and Video). For each temporal level, evaluators are required to rate the grade of dynamics from "static" to "very high dynamics". To guide the annotation process, we provide specific prompts for each temporal level. [4]. We conduct between-group correlation analyses using Pearson's correlation, Kendall's correlation, and the win ratio to evaluate the consistency of dynamics scores with respect to human ratings.

The evaluation of the naturalness metric follows the same process, where a higher human assigned grade indicates a greater degree of naturalness.

**Evolution**. We calculate dynamics grades and naturalness for generated videos on the proposed DEVIL benchmark. For dynamics scores at multiple temporal levels, we integrate them using the linear regression model defined by Eq. 12. For each linear regression model, it takes the human evaluation results as ground-truths, trained upon 75% of the randomly selected videos and tests on the remaining 25% videos. During testing, the human alignment performance is reflected by the correlation $e.g.$, Pearson and Kendall correlation coefficients and win ratio, between predicted and human-evaluated dynamics grades. The win ratio involves comparing each video against others with different grades of dynamics. For instance, a video rated as "high dynamics" by evaluators should score lower in dynamics than any video rated as "Very high dynamics" but higher than those rated as "static".

Table 3 shows the assessment results of the six T2V models. It can be seen that the dynamics metrics and the naturalness metric exhibit a strong alignment with human evaluation. The improved metrics ($\mathbf{D}_f$, $\mathbf{D}_s$, $\mathbf{D}_v$ defined in Sec. 3.3) further enhance the alignment with human evaluations.

Table 3: Human alignment by correlation between dynamics scores and human ratings on the proposed DEVIL benchmark. Video generation is based on text prompts in DEVIL. "PC" denotes Pearson's correlation, "KC" Kendall's correlation, and "WR" the win ratio.

| Scores | | PC ↑ | KC ↑ | WR ↑ |
|---|---|---|---|---|
| Inter-frame | $S_{ofs}$ | 93.1 | 89.9 | 79.2 |
| | $S_{sd}$ | 91.7 | 88.0 | 78.1 |
| | $S_{pd}$ | 96.4 | 93.2 | 86.1 |
| | $S_f$ | **96.5** | **93.5** | **86.5** |
| Inter-segment | $S_{pa}$ | 95.1 | 94.3 | 87.0 |
| | $S_g$ | 94.6 | 93.0 | 85.6 |
| | $S_s$ | **95.8** | **94.8** | **87.7** |
| Video level | $S_{te}$ | 96.4 | 93.7 | 83.5 |
| | $S_{tsd}$ | 97.7 | 96.4 | 90.5 |
| | $S_v$ | **98.0** | **97.2** | **91.4** |
| Naturalness | | 79.0 | 75.5 | 52.4 |

## 5.2 Influence of Frame Rate.

---

[4]Please refer to Appendix G for details

Table 6: Evaluation of existing metrics and improved metrics. These metrics include **M**otion **S**moothness (MS), **B**ackground **C**onsistency(BC), **S**ubject **C**onsistency(SC), and **Nat**uralness (Nat).

| Model | Existing Metrics | | | | Improved Metrics | | | |
|---|---|---|---|---|---|---|---|---|
| | MS | BC | SC | Nat | MS | BC | SC | Nat |
| GEN-2 [2] | 99.4 | 97.2 | 95.6 | 81.6 | 57.9 | 55.6 | 53.3 | 38.0 |
| Pika [4] | 99.4 | 96.5 | 93.2 | 69.0 | 57.7 | 55.8 | 53.1 | 32.4 |
| VideoCrafter2 [14] | 97.7 | 97.4 | 95.5 | 70.8 | 48.1 | 47.7 | 46.5 | 33.1 |
| OpenSora [23] | 95.4 | 94.3 | 88.7 | 63.6 | 69.9 | 69.5 | 63.1 | 37.1 |
| StreamingT2V [19] | 94.9 | 91.0 | 85.2 | 55.2 | 70.7 | 68.0 | 61.2 | 33.2 |
| FreeNoise-Lavie [31] | 95.5 | 94.1 | 90.1 | 73.4 | 77.4 | 68.1 | 76.3 | 51.4 |
| VideoCrafter1 [13] | 95.8 | 95.3 | 92.8 | 75.4 | 61.3 | 60.7 | 57.9 | 39.7 |
| Hotshot-XL [3] | 97.0 | 95.5 | 93.4 | 85.6 | 55.7 | 54.7 | 52.1 | 45.1 |
| Show-1 [48] | 97.1 | 94.4 | 91.5 | 74.4 | 62.9 | 61.8 | 58.6 | 43.0 |
| ModelScope [40] | 95.8 | 93.5 | 89.3 | 71.6 | 70.5 | 68.2 | 62.6 | 44.9 |
| ZeroScope [5] | 98.2 | 94.8 | 89.3 | 74.6 | 49.1 | 46.8 | 45.6 | 34.2 |

Table 5 demonstrates how frame rate influences the correlation between dynamics scores and human evaluations. Experiments indicate that our dynamics scores maintain a high correlation (>0.9) with human ratings across various frame rates. To mitigate the impact of frame rate variations on dynamics, we standardized the frame rate of each video to 8 FPS.

Table 5: Influence of frame rate on the consistency of dynamics scores with human evaluation (measured by Pearson's correlation).

| Dynamics | 4FPS | 8FPS | 16FPS | Origin |
|---|---|---|---|---|
| Inter-frame | 0.952 | 0.950 | 0.946 | 0.951 |
| Inter-Segm | 0.952 | 0.954 | 0.954 | 0.953 |
| Video | 0.967 | 0.967 | 0.967 | 0.967 |

### 5.3 Computation Efficiency

Our dynamics metrics offer high computational efficiency, achieving around 10 frames per second on a single NVIDIA A100 GPU, and are scalable to multiple GPUs.

### 5.4 Evaluation of Video Dynamics

We evaluate the dynamics range $M_{range}$, and dynamics controllability $M_{control}$ of T2V models on the proposed DEVIL benchmark. We also evaluate dynamics ranges at different temporal scales: inter-segment dynamics range $M_{range}^f$, inter-segment dynamics range $M_{range}^s$, and video level dynamics range $M_{range}^v$. The results are shown in Table 4. The GEN-2 [2] and Pika [4] models score high in dynamics controllability but low in range due to their generation of low-dynamic videos. Conversely, the FreeNoise-Lavie method [31] attains high range but low controllability, suggesting it produces videos with dynamics that do not align well with text prompts.

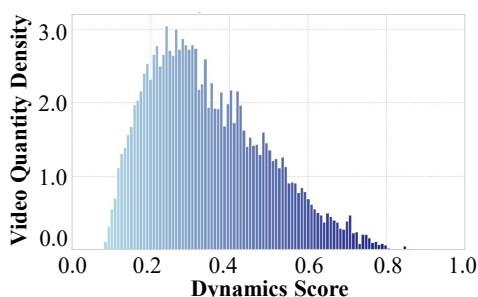

Figure 5: Video quantity density $w.r.t.$ dynamics score of the WebVid-2M dataset.

### 5.5 Improved Evaluation Metrics

As shown in Table 6, the existing metrics exhibit an obvious negative correlation when embedded with dynamics, indicating that these models can achieve high scores on these metrics by generating low-dynamic videos rather than high-quality content.

### 5.6 Insights from Video Dynamics Analysis

**Biased Dynamics Distribution of Existing Dataset.** The distribution of dynamics of the video datasets (such as WebVid2M [8]) is biased. The statistical result is shown in Fig. 5. It can be seen that most of the videos have a small dynamics score ($\leq 20$). The limited number of videos with high dynamics scores restricts the model's ability to generate dynamics-rich videos which are common in

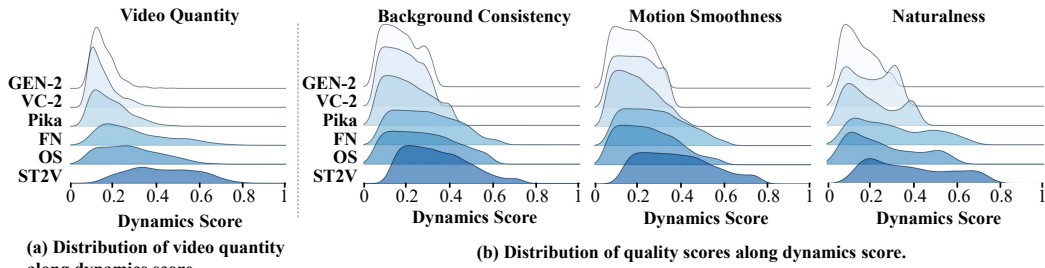

**(a) Distribution of video quantity along dynamics score.**

**(b) Distribution of quality scores along dynamics score.**

Figure 6: Distributions of video quantity and quality scores along the dynamics score for various video generation models including: GEN-2 [2], Pika [4], VideoCrafter2(VC-2) [14], Open-Sora(OS) [23], StreamingT2V [19] and FreeNoise-Lavie(FN) [31]. Subplot (a) shows video quantity distribution. Subplots (b) display the distribution of quality score of generated videos in terms of Background Consistency, Motion Smoothness, and Naturalness, respectively. All videos are generated based on our text prompt benchmark.

practical applications. Therefore, existing datasets should be expanded in terms of dynamics, and the proposed metrics can provide guidance for this expansion.

**Prompt-Video Bias of Existing Datasets.** We used the dynamics controllability metric to evaluate two popular datasets, WebVid2M [8] and MSR-VTT [46], by using the ground-truth text prompts and videos. Unfortunately, they respectively achieve alignment scores of 36.31 and 52.98. The poor performance indicates that the two datasets can not provide sufficient information/guidance while training the video generation models. To train better video generation models, the text prompt of these datasets requires to be elaborated on aspects of dynamics.

**Limited Real-World Simulation Ability of Existing Methods.** As shown in Fig. 6, we performed a statistical analysis of frequency, visual quality, motion smoothness and naturalness metric scores for SOTA methods based on the distribution of dynamics score. When the dynamics score is low, videos generated by these SOTA models perform well across the four metrics mentioned. As the dynamics score rises, these metrics, particularly naturalness, tend to decrease significantly. This decline may be due to the models' focus on optimizing the generation of simple, slow-motion content, with dynamics not considered in the evaluation metrics.

# 6 Conclusion

We proposed DEVIL, a comprehensive and constructive evaluation protocol for T2V generation models. In the protocol, we defined a set of dynamics metrics corresponding to multiple temporal granularities, and a new benchmark of text prompts under multiple levels of dynamics. Based on the distribution of dynamics scores over the benchmark, we assessed the generation capacity of T2V models, characterized by dynamic ranges and degree of T2V alignment. Experiments show that DEVIL enjoys 90% consistency with human evaluation results, demonstrating the potential to be a powerful tool for advancing T2V generation models.

**Limitations.** At present, the grades of dynamics remain limited, which should be improved to more fine-grained grades. Furthermore, only a limited number of T2V models are evaluated using the proposed protocol. A more comprehensive evaluation of T2V models should be done in future work.

# 7 Acknowledgment

This work was supported by National Natural Science Foundation of China (NSFC) under Grant 62472402 and 62225208, and the Fundamental Research Funds for the Central Universities. This work was also supported by the Centre for Augmented Reasoning, an initiative by the Department of Education, Australian Government.

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

# Appendix

## A   Inter-segment Dynamics

Inter-segment dynamics quantifies the diversity of patterns between video segments by measuring the similarity of features across different segments. In this section, we investigate the influence of various factors on inter-segment dynamics, including the method for video segmentation, the proportion ratio $r$, and the overall length of the video.

**How to segment videos?** In the computation of global aperiodicity within inter-segment dynamics, video segmentation is essential. Table 7 compares the effects of proportional video segmentation and keyframe-based segmentation on the performance of inter-segment dynamics. We observe that both segmentation methods achieve comparable levels of correlation with human evaluations, with Pearson's correlation coefficients of 0.96 and 0.95, and Win Ratios of 0.85 and 0.87, respectively. Given that proportional video segmentation facilitates the simultaneous comparison of videos of varying lengths and the frequency of pattern changes, we have opted to utilize proportional video segmentation in our implementation of global aperiodicity.

Table 7: Comparison of keyframe-based and proportional video segmentation methods

| Video Segment Method | Pearson's Correlation | Kendall's Correlation | Win Ratio |
|---|---|---|---|
| Key-frame-based | 0.96 | 0.94 | 0.85 |
| Proportional | 0.95 | 0.93 | 0.87 |

**Influence of proportional factor $r$.** Table 8 illustrates the impact of the parameter $r$ on the performance of inter-segment dynamics. When $r$ is set at $1/8$, $1/4$, and $1/2$, the Pearson's correlation coefficients with human ratings are 0.92, 0.94, and 0.93 respectively. These results indicate that inter-segment dynamics is robust to variations in $r$. Ultimately, we selected $r = 1/4$ as it achieved the highest correlation with human evaluations.

Table 8: Influence of the proportion factor $r$ on the performance of inter-segment dynamics

| Proportion Factor $r$ | Pearson's Correlation | Kendall's Correlation |
|---|---|---|
| $1/8$ | 0.92 | 0.90 |
| $1/4$ | 0.94 | 0.91 |
| $1/2$ | 0.93 | 0.90 |

**Influence of Video Length.** In Table 9 .We group videos based on the video length (max is 8s in tested models) and study the relation between dynamics scores and human scores. Inter-segment dynamics robustly achieves over 90% correlation whatever the video length is.

Table 9: Influence of video length the performance of inter-segment dynamics.

| Video Length (s) | Pearson's Correlation | Kendall's Correlation |
|---|---|---|
| 2 | 0.96 | 0.94 |
| 4 | 0.93 | 0.91 |
| 8 | 0.94 | 0.90 |

## B   Model Weight of Human Alignment Module

Table 10 shows the weights of each dynamics score in the human alignment module.

Table 10: Weights of each dynamics score in the human alignment module.

| Temporal Scale | Dynamics Score | Typical Value | Weight |
|---|---|---|---|
| Inter-frame | $D_{ofs}$ | 62.00 | 6.70E-04 |
| | $D_{sd}$ | 1.00 | 0.17 |
| | $D_{pd}$ | 33.00 | 0.03 |
| Inter-segment | $D_{pa}$ | 0.20 | 2.20 |
| | $D_{ga}$ | 0.80 | 0.63 |
| Video | $D_{te}$ | 7.00E+04 | 1.00E-05 |
| | $D_{tsd}$ | 0.20 | 1.46 |

Table 11: Pearson correlation coefficient between the dynamics metrics and the existing metrics including aesthetic score [45], technical score [45] visual quality [45], motion smoothness [24], subject consistency [24] and background consistency [24] and our naturalness.

| | Aesthetic Score | Technical Score | Visual Quality | Motion Smoothness | Subject Consistency | Background Consistency | Naturalness |
|---|---|---|---|---|---|---|---|
| GEN-2 [2] | -0.19 | -0.09 | -0.12 | -0.54 | -0.88 | -0.73 | -0.50 |
| Pika [4] | -0.40 | -0.20 | -0.28 | -0.65 | -0.88 | -0.78 | -0.47 |
| VideoCrafter2 [14] | -0.25 | -0.20 | -0.24 | -0.59 | -0.87 | -0.76 | -0.36 |
| OpenSora [23] | -0.20 | -0.27 | -0.26 | -0.70 | -0.90 | -0.83 | -0.43 |
| StreamingT2V [19] | -0.15 | -0.21 | -0.23 | -0.57 | -0.89 | -0.81 | -0.36 |
| FreeNoise-Lavie [31] | -0.37 | -0.31 | -0.35 | -0.75 | -0.91 | -0.86 | -0.48 |
| Average | -0.26 | -0.21 | -0.25 | -0.63 | -0.81 | -0.79 | -0.43 |

## C   Correlation Between Existing Metrics and Dynamics

In Section 4, to identify the relevance between existing metrics with the dynamics metrics, we provide a bi-variate analysis strategy. Based on bi-variate analysis, we provide detailed correlation results for the models. In Table 11, the Pearson correlation coefficients between the dynamics scores and existing metrics, including aesthetic score, technical score, visual quality, motion smoothness, subject consistency, background consistency, and naturalness, are detailed.

The results indicate a clear trade-off between video dynamics and various existing metrics in T2V models. As dynamic complexity increases, there tends to be a decline in motion smoothness, subject consistency, background consistency, and naturalness. The aesthetic, technical, and visual quality metrics show relatively low correlation, which can be attributed to the fact that these metrics evaluate video frames independently, ignoring temporal relationships between frames.

## D   Comprehensive Evaluation Metrics

Let $S^{(i)}$ denote a score of generated video $i$. Existing metrics simply average the scores of all videos to obtain the metric score $S$ of the $T2V$ model:

$$S = \frac{1}{|T|} \sum_{i=1}^{|T|} S^{(i)}, \qquad (15)$$

where $|T|$ is the total number of generated videos. Considering that some existing metrics show a considerable negative correlation with the video's dynamics score, they fail to prevent models from generating low-dynamic videos.

To address this issue, we enhance existing metrics by integrating human-aligned dynamics scores, preventing models from attaining high scores by producing low-dynamic videos. Specifically, we first equally divide the human-aligned dynamics score into $L = 13$ intervals. We then calculate the mean scores $S_l$ at each interval $l$. The improved metric $S^*$ is defined as the average of $S_l$ across all intervals:

$$S^* = \frac{1}{L} \sum_{l=1}^{L} S_l. \qquad (16)$$

# E    Assigning Dynamics Grades to Text Prompts

As described in Section 3.1, we collect approximately 50,000 text prompts from existing benchmarks, including 19 object categories and 4 scene categories. Using GPT-4 coarse classification and human refinement, we construct the DEVIL prompt benchmark. The process of categorizing dynamics grades using GPT-4 is illustrated in Figure 7. In specific, we instruct GPT-4 to perform classification on the rate of content change. To enhance GPT-4's classification accuracy, we further provide detailed criteria and examples for each dynamics grade. In the post-processing step, we recruit six human annotators to refine the dynamics grades over three months. Finally, we sample about 800 text prompts at different dynamics grades to ensure a uniform distribution across the grades.

# F    Details of Naturalness

We employed the advanced multi-modal large model, Gemini-1.5 Pro [1], equipped with video understanding capabilities, to assess and classify the naturalness of video content. As shown in Fig. 8, we demonstrate the process through which the model analyzes videos and assigns naturalness ratings. The figure details the five different levels used to evaluate video naturalness, ranging from "Completely Fantastical" to "Almost Realistic". Each level is defined based on how closely the video content aligns with the real world. Additionally, the figure includes two examples of video evaluations: the first video is rated as "Almost Realistic" due to its high conformity with reality, while the second video, due to minor distortions—such as the unrealistic number of legs on a dog—is rated as "Slightly Unrealistic". These examples validate the plausibility of the proposed naturalness metric.

# G    Human Annotation

To align human evaluations with automated metrics, we annotated a series of videos generated by SOTA T2V models. We initiated the process by generating videos using prompts from the DEVIL benchmark with six advanced T2V models including GEN-2, Pika, VideoCrafter2, OpenSora, StreamingT2V, and FreeNoise-Lavie. Subsequently, we developed a video annotation toolbox for evaluating the dynamics and naturalness of videos. As shown in Figure 9, the toolbox allows annotators to assess the dynamics of the videos across five grades, from almost static to very high dynamics, and the naturalness from almost real to completely unreal. To guarantee high-quality and consistent evaluations, we recruit six annotators who have undergraduate degrees and provided them with detailed training.

# H    Visual comparison

In Section 3, we use text prompts with different dynamics grades to generate videos with T2V models. Here, we provide visual results of the generated videos.

Given the provided text, classify each text segment according to the scene and background dynamics using the following criteria. For each text segment, inherit the serial number at the beginning of the text and a classification label from the list below.

**Classification Criteria:**

    **Almost Static**: Minimal changes in scene or background, almost static.

    **Example**: "A room where only the fading light changes."

    **Low Dynamics**: Slow and slight changes in scene, usually slow motion.

    **Example**: "A balloon slowly rising, with a focus on its details."

    **Medium Dynamics**: Noticeable activity and changes, but relatively smooth overall.

    **Example**: "A child and dog moving from grass to sand."

    **High Dynamics**: Fast actions and changes.

    **Example**: "A chase scene with rapid transitions and complex maneuvers."

    **Very High Dynamics**: Extremely rapid and frequent video content changes.

    **Example**: "A battle scene with quick cuts and intense action."

Instructions:

    For each section of text, assign a dynamics grade classification based on the provided criteria. List the serial number inherit from the beginning of the text followed by the classification.

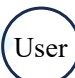

1.    A car drifts sharply around a corner, almost hitting a bystander.
2.    little girl putting down and picking up her bear plush.
3.    the dead bird is on the ground.
…

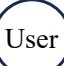

1.    High Dynamics
2.    Low Dynamics
3.    Almost Static
…

Figure 7: Illustration of prompt coarse categorization using GPT-4 [30].

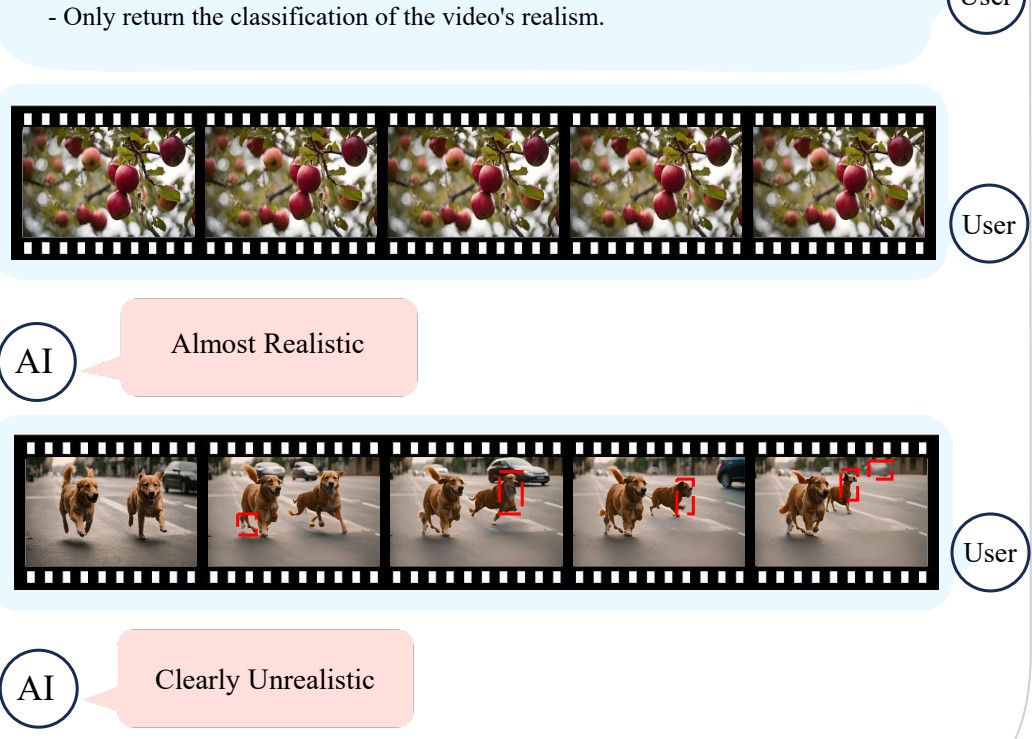

Figure 8: Illustration of naturalness calculation for generated videos using Gemini-1.5 Pro [1].

Enter video index to start.

0

Load Video

Previous

Next

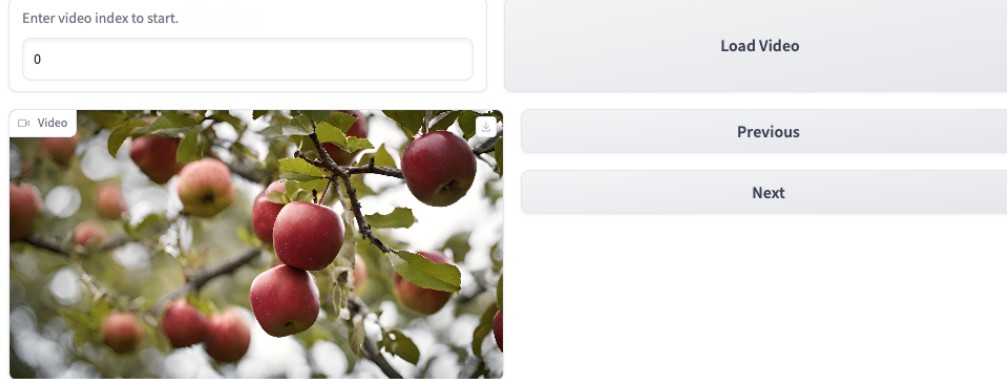

**Question 1: Which level of inter-frame change does this video belong to?**

1. **Almost Static**: The scene is almost completely still, or there is only minimal detail change, such as a person standing still or an almost static sky.

2. **Low Dynamics**: There are slow movements or scene changes, such as a person walking slowly or a slowly shifting viewpoint.

3. **Medium Dynamics**: There are moderate-speed movements or scene changes, such as a person walking at a normal pace or normal wind and rain.

4. **High Dynamics**: There are fast movements or scene changes, such as a person running.

5. **Very High Dynamics**: There are very fast movements and scene transformations between frames, such as vehicles speeding or significant scene content changes.

| Almost Static: 1 | Low Dynamics: 2 | Medium Dynamics: 3 | High Dynamics: 4 | Very High Dynamics: 5 |

**Question 2: Which level of inter-segment dynamic variation or repetition does this video belong to?**

1. **Almost Static**: The video is almost repetitive with only minor changes, such as nearly static objects.

2. **Low Dynamics**: The video content is almost repetitive with slight variations, such as leaves shaking.

3. **Medium Dynamics**: Some content in the video recurs, but there are considerable changes or one major change in background content. For example, changes in pedestrian or traffic flow.

4. **High Dynamics**: The video content contains little repetition, with unpredictable movement patterns of objects or significant changes in background content. For example, movements of players in a sports match.

5. **Very High Dynamics**: Content is almost non-repetitive, with extremely complex and diverse object and scene changes. For example, tightly edited action sequences or live broadcasts of high-speed races.

| Almost Static: 1 | Low Dynamics: 2 | Medium Dynamics: 3 | High Dynamics: 4 | Very High Dynamics: 5 |

**Question 3: Which level of global dynamics does this video belong to?**

1. **Almost Static**: Almost no changes, with only minimal motion or environmental changes.

2. **Low Dynamics**: Slight changes or slow movements, very little object variation, and limited viewpoint changes.

3. **Medium Dynamics**: Noticeable changes in objects or background, significant movement or changes in quantity and appearance of objects, the background is completely different but still depicts the same scene.

4. **High Dynamics**: Quick changes in objects and scenes, and rich actions.

5. **Very High Dynamics**: Significant changes in objects, dramatic increase or decrease in quantity or frequent changes in appearance, very rapid background changes, frequent scene and viewpoint switches, and changes in the depicted scenes.

| Almost Static:1 | Low Dynamics:2 | Medium Dynamics:3 | High Dynamics:4 | Very High Dynamics:5 |

**Question 4: What is the level of naturalness in this video's content?**

○ **Completely Unreal**: Completely detached from reality across all times and scales, filled with fantasy or surreal elements.

○ **Clearly Unreal**: Long-term or macro-scale distortions are significant, leading to an overall departure from reality. Entire scenes are unrealistic, defy physical laws, or large objects are unreal.

○ **Moderately Unreal**: There are clear distortions on a short-term or meso-scale, but the overall plot remains relatively coherent. For example, medium-sized objects or scenes are unrealistic.

○ **Slightly Unreal**: Distortions occur only instantaneously or on a micro-scale, and are not easily noticeable. Such as unnatural facial expressions of characters or unnatural scene textures.

○ **Almost Real**: No noticeable distortions, completely consistent with reality.

| Completely Unreal: 1 | Clearly Unreal: 2 | Moderately Unreal: 3 | Slightly Unreal: 4 | Almost Real: 5 |

Figure 9: Toolbox for dynamics and naturalness annotation.

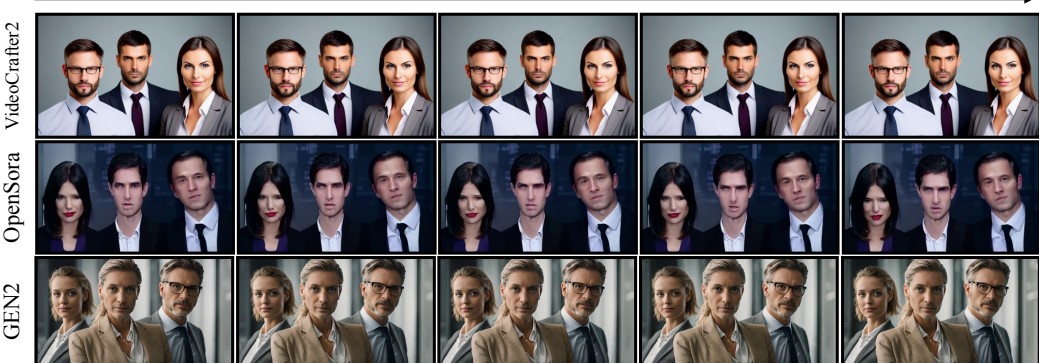

"Text Prompt": "Portrait of three business people looking at the camera."
"Dynamic grade": "Static"

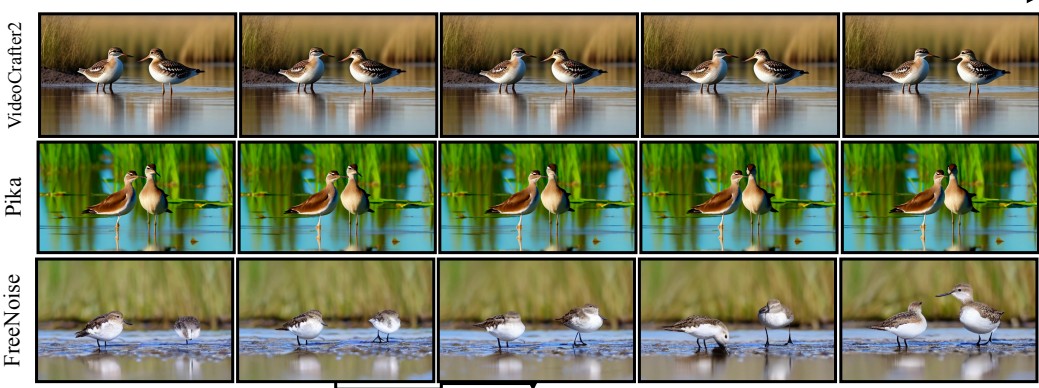

"Text Prompt": "Tringa glareola. two wood sandpipers in the summer. standing on land near the lake in the north of siberia."
"Dynamic grade": "Low"

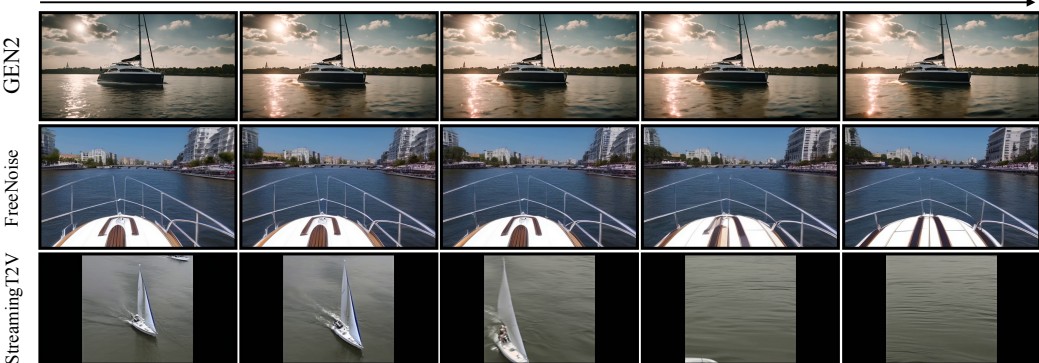

"Text Prompt": "Yachting on wide city river. outdoor activities, summer vacation"
"Dynamic grade": "Medium"

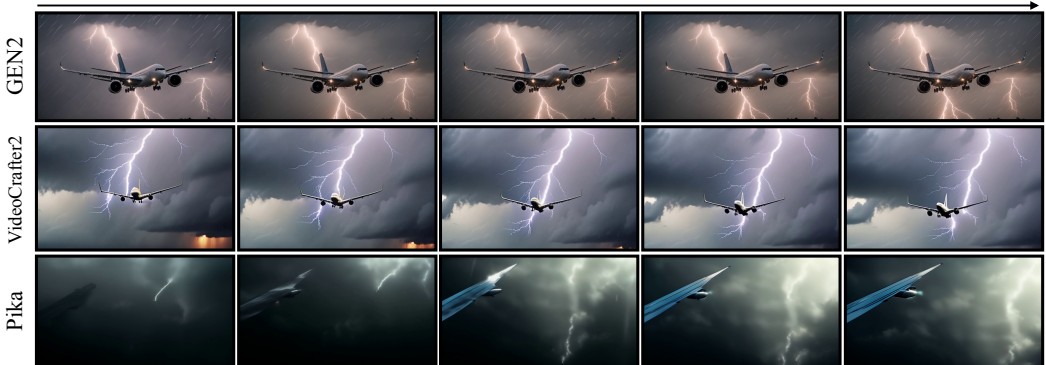

"Text Prompt": "Commercial airplane dodging lightning during a turbulent storm, rain-drenched windows."
"Dynamic grade":  "High"

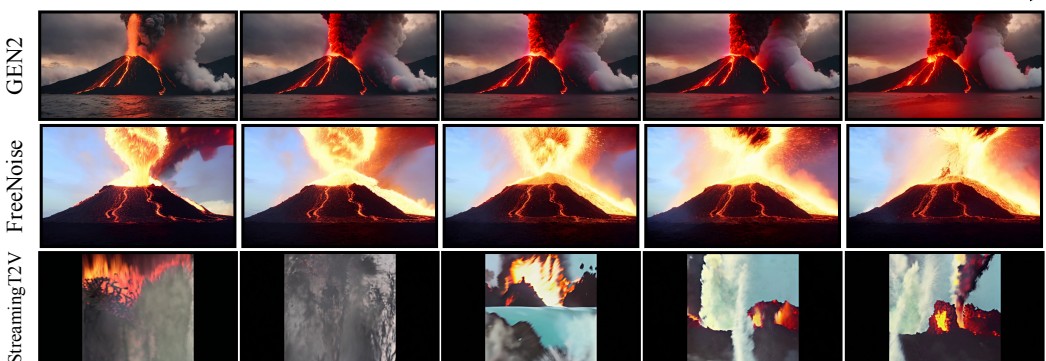

"Text Prompt": "High-speed shots of a volcanic eruption engulfing a tropical island, with lava fountains
spewing molten rock and the environment transforming from idyllic paradise to hellish
landscape of ash and fire."
"Dynamic grade":  "Very High"

