# OpenReview forum: "Evaluation of Text-to-Video Generation Models: A Dynamics Perspective"
_NeurIPS.cc/2024/Conference — NeurIPS 2024 poster_

### Official Review · Reviewer_Kt6U · 2024-06-24

**Soundness:** 2
**Presentation:** 2
**Contribution:** 2
**Rating:** 5
**Confidence:** 3

**Summary:**

This paper proposes a new evaluation metric for the text-to-video model, and this metric in particular focus on the dynamics on the generated video. Their metric is based on three sub-scores: inter-frame dynamics score, inter-segment dynamics score, and video-level dynamics. They did

**Strengths:**

It explores a more fine-grained protocol to evaluate the dynamics of the T2V generation.

They have done extensive evaluation on the exiting T2V models, and obtained some insights.

They propose the "improved metric", which would give higher weight to the large-dynamic range compared to the "existing metric". It would be useful to detect whether the generated videos of a model cover all dynamic ranges.

**Weaknesses:**

I feel this paper over-emphasizes existing evaluation works ignore the dynamics, this is not quite what I saw. I think most of papers already paid attention to the dynamics evaluation, they are just not so detailed and fine-grained as in this paper. For example, EvalCrafter has Motion Quality section for considering the dynamics.

I think there are already some existing metric to quantify the dynamics in the video, and they're simpler, such as Motion Quality in EvalCrafter. I think there should be a direct comparison between the proposed metric and the existing dynamics metric. Maybe the authors can consider to use the average of three motion quality metrics in EvalCrafter (after normalization) and show which one is aligned with human evaluation better?

Another problem is for the Inter-segment Dynamics Score, I think the way to take segments is too brutal-force and sketchy, for example, just take every 8 frames. This maybe totally misaligned with the actual semantic segments, in that case, I feel the Inter-segment Dynamics Score does not make much sense.

For the naturalness metric's definition, why can we trust Gemini-1.5pro? Authors may want to justify this.

**Questions:**

- Why is this protocol called DEVIL?

- What are the cross and circles in the Dynamics Evaluation plot in Figure 2? I'm confused what this plot wants to convey.

- Figure 3 may need to be improved. (b) is only the word cloud for DEVIL instead of for all three benchmarks.

- The reference in 136 line seems to be incorrect, it is not a paper for optical flow.

- What's SIM in equation (2)?

- What does I function mean in equation (14)?

Typo:

Line 223: overall h??

**Limitations:**

.

---

> ### Author Rebuttal · Authors · 2024-08-07
>
> #### **Q1: Compare to existing metrics, such as Motion Quality in EvalCrafter. Consider to use the average of three motion quality metrics in EvalCrafter (after normalization) and show which one is aligned with human evaluation better?**
> The differences between our method and the existing metrics, e.g., Motion Quality in EvalCrafter, are as follows:
> 1.  Dynamics Assessment
> - Vbench and EvalCrafte rely solely on optical flow, limiting analysis to inter-frame pixel motion dynamics. They fail to capture non-pixel-motion dynamics, such as variations in illumination, style, and color.
> - Our method employs seven dynamic indicators across three temporal scales (inter-frame, inter-segment, and full-video), comprehensively assessing video dynamics. This approach significantly improves correlation with human evaluation, when evaluating inter-frame dynamics it achieving an 84% win ratio, surpassing the 73% of optical flow.
> 2. Evaluation of Models' Capabilities on Dynamics
> - Vbench and EvalCrafter created Dynamic Degree and Flow Score metrics based on optical flow, both favoring videos with greater dynamics. However, they overlook the need for videos to match varying dynamics in text prompts, such as low dynamics for low-dynamics descriptions and high dynamics for high-dynamics descriptions.
> - Although EvalCrafter's Motion AC-Score assesses a model's ability to generate videos with high or low dynamics, it is a binary metric, which is a coarse measurement and can not reflect the true dynamic controllability.
> - Instead, our Dynamics Range, $M_{range}$, measures a model's ability to generate videos with various dynamics, and Dynamics Alignment, $M_{align}$ measures how well models can align video dynamics to text prompts across five grades, enabling a more reasonable and precise assessment of a model's capabilities on dynamics.
> 3. Quality Metrics
> - Vbench and EvalCrafter assess video dynamics and quality separately, neglecting their correlation. It results in that the models tend to generate low-dynamics videos to ensure high quality, limiting to reflect the model quality across different dynamics.
> - EvalCrafter's Action Recognition is limited to evaluating human actions using recognition scores and doesn't directly evaluate video quality across different dynamics.
> - By incorporating dynamics into existing quality metrics, we quantitatively assess a model's ability to generate high-quality videos across dynamic ranges, achieving a more comprehensive evaluation.
>
> Following your suggestion, we use the average of motion quality metrics in EvalCrafter (after normalization) and calculated the correlation of this average metrics with human evaluation. It obtains 83% Pearson's correlation andb 74% win ratio, which are significantly lower than those in our proposed dynamics metric ( 95% Pearson's correlation and an 84% win ratio).
>
> #### **Q2: Inter-segment dynamics Score**
> We would like to clarify that the length of each segment is not 8 frames. Instead, segments are divided proportionally according to the video's duration when evaluating global aperiodicity. Experimental results indicate that the global aperiodicity in inter-segment dynamics is robust with respect to the proportion, maintaining a high correlation with human evaluations (>90%) when assessed at proportions of 1/8, 1/4, and 1/2 of the video's length.
>
> | Proportion |  PC  |  KC  |
> |----------|----|----|
> |    1/8     | 0.92 | 0.90 |
> |    1/4     | 0.94 | 0.91 |
> |    1/2     | 0.93 | 0.90 |
>
> Based  the on the observation that different patches within the video take different lengths of time to change, we assess the correlation of changes over different time interval (from 1 to the full video length) base on the temporal autocorrelation factor in this paper. This constructs the Inter-segment Dynamics feature combining with the global aperiodicity score, which achieves a 95% correlation with human judgments.
>
> #### **Q3: Why trust Gemini 1.5 Pro for Naturalness Evaluation**
> 1. Gemini 1.5 Pro has been validated to have the best video comprehension capabilities among Multimodal Large Language Models (MLLM) currently available [a].
> 2. Our validation indicates that Gemini 1.5 Pro achieves a 78% correlation with human ratings in terms of naturalness as shown in Table 3 in the paper.
>
> [a] Fu C, Dai Y, Luo Y, et al. Video-MME: The First-Ever Comprehensive Evaluation Benchmark of Multi-modal LLMs in Video Analysis[J]. arXiv preprint arXiv:2405.21075, 2024.
>
> #### **Q4: Why is this protocol called DEVIL?**
> "DEVIL" stands for “**D**ynamics-based **E**valuation of **VI**deo generation mode**L**s”. The name also hints at the tough challenges that dynamics bring to video generation models.
>
> #### **Q5: What are Cross and circles in Figure 2**
> We apologize for any confusion. The rightmost plot in Figure 2 illustrates the calculation process for Dynamics Alignment and Dynamics Range. A circle represents videos where the dynamics are correctly aligned with the prompt, while a cross indicates videos where the dynamics are misaligned.
>
> #### **Q6: Figure 3 may need to be improved. (b) is only the word cloud for DEVIL instead of for all three benchmarks.**
> Thanks for pointing this out. It is true that Figure 3 (b) is the DEVIL's word cloud instead of all three benchmarks. We will rectify the caption.
>
> #### **Q7: Wrong reference in line 136**
> Thank you for pointing out the error. We have updated the reference to:
> Teed, Zachary and Jia Deng. "RAFT: Recurrent All-Pairs Field Transforms for Optical Flow." European Conference on Computer Vision (2020).
>
> #### **Q8: What's SIM in equation (2)?**
> SIM in equation (2) represents the cosine similarity between two feature vectors.
>
> #### **Q9: What does I function mean in equation (14)?**
> The I in Equation (14) refers to the indicator function.
>
> #### **Q10: Line 223: overall h???**
> Thank you for pointing out the typo. The correct sentence should read:
> "The overall naturalness is then determined by averaging the scores of all videos."

---

> > ### Comment · Reviewer_Kt6U · 2024-08-12
> >
> > I'll raise my score but I'm still a bit uncertain on the segment way of Inter-segment Dynamics.

---

> > > ### Author Response · Authors · 2024-08-13
> > >
> > > Thank you for recognizing our work and rebuttal.
> > >
> > > Inter-segment Dynamics measures the dynamics of temporal patterns by assessing the correlation/similarity between video segments.
> > > Proportional video segmentation allows for the simultaneous comparisons of videos with varying lengths while taking into account the frequency of pattern changes.
> > > In our rebuttal, we have demonstrated that Inter-segment Dynamics is robust to segment ratios.
> > > Additionally, we compared it with keyframe-based segmentation methods, as shown below:
> > >
> > > | Segment method| Pearson Correlation    | Kendall Correlation    |  Win Ratio    |
> > > |---------------|:-------:|:-------:|:-------:|
> > > | Keyframe-based| 0.963 | 0.940 | 0.848 |
> > > | Ratio         | 0.954 | 0.934 | 0.865 |
> > >
> > > Our method is comparable to keyframe-based segmentation.
> > > In the future, we will continue to explore better methods for calculating inter-segment dynamics.
> > > We appreciate your consideration and hope for a favorable review.
> > >
> > > Thank you so much!

---

> > > > ### Comment · Reviewer_Kt6U · 2024-08-13
> > > >
> > > > I'm a bit confused, why win rate and correlation are conflicting with each other?

---

> > > > > ### Author Response · Authors · 2024-08-13
> > > > >
> > > > > Given that humans can only group videos into groups based on their dynamics rather than precisely quantifying each video's dynamics, we conduct between-group analysis when calculating the win ratio and correlation.
> > > > >
> > > > > - **Correlation metrics**  calculate the between-group correlations, focusing on overall trends.
> > > > >
> > > > > - The **Win Ratio** focuses on the accuracy of pairwise comparisons, providing a more fine-grained level of analysis.
> > > > >
> > > > > Overall, scores on all metrics are high.
> > > > > The slight differences observed between correlation and win ratio can be attributed to their distinct calculation methods.
> > > > >
> > > > > We hope that all the rebuttals can solve your concerns towards positive rating.

---

> > > > > > ### Comment · Reviewer_Kt6U · 2024-08-14
> > > > > >
> > > > > > Thanks! I have raised the score. Please include these discussions in the revised paper.

---

> > > > > > > ### Author Response · Authors · 2024-08-14
> > > > > > >
> > > > > > > We greatly appreciate you raising the score. As you recommended, we will include these discussions in the revised version.

---

> > > ### Author Response · Authors · 2024-08-13
> > >
> > > Before the end of the rebuttal, we welcome further discussion and hope for further raising of the rating. We would like to appreciate your further encouragement. Thank you very much.

---

> ### Comment · Area_Chair_gdzK · 2024-08-12
> **Final thoughts**
>
> Hi Reviewer Kt6U,
>
> The discussion period is ending soon. Please take a moment to review the author's rebuttal and share any remaining concerns or questions.
>
> Thank you,
> AC

---

### Official Review · Reviewer_VYfN · 2024-07-07

**Soundness:** 3
**Presentation:** 3
**Contribution:** 3
**Rating:** 6
**Confidence:** 3

**Summary:**

Effective evaluation protocols are essential for developing advanced text-to-video (T2V) generation models. Current protocols primarily address temporal consistency and content continuity but often neglect the dynamics of video content, which are crucial for visual vividness and fidelity to text prompts. This study introduces DEVIL, a protocol that emphasizes dynamics by defining metrics across multiple temporal granularities and creating a benchmark of text prompts graded by dynamics levels. DEVIL evaluates T2V models using metrics of dynamics ranges and T2V alignment, enhancing existing metrics from a dynamics perspective. Experimental results show that DEVIL achieves up to 90% consistency with human evaluations, demonstrating its potential as a powerful tool for advancing T2V generation models.

**Strengths:**

1. **Introduction of New Dynamics Metrics**: The study presents a novel set of dynamics metrics that enhance the evaluation of text-to-video (T2V) generation models. These metrics focus on the dynamics dimension, which assesses the visual vividness and fidelity of video content to text prompts across multiple temporal granularities.

2. **Comparative Analysis with Existing Methods**: The research includes a thorough comparative analysis of existing evaluation methods, highlighting their limitations in addressing video dynamics. By juxtaposing these methods with the newly proposed dynamics metrics, the study demonstrates how the latter offers a more detailed assessment of T2V models.

3. **Code Submission for Reproducibility**: The authors have submitted the code used in their study. This openness is crucial for validating the proposed methods and fostering innovation in the community.

**Weaknesses:**

1. **Difficult to Assess Dynamics from Images Alone**: Evaluating the impact of dynamics solely through images is challenging. To better understand the effectiveness of the proposed dynamics metrics, it would be beneficial to provide several real videos along with their corresponding scores. This would allow for a more accurate assessment of how well the metrics reflect the dynamics and visual vividness of real video content. Without these examples, it is hard to gauge the practical utility and accuracy of the dynamics scores.

2. **Comparison of Existing T2V Methods**: Similarly, assessing the dynamics of generated videos from existing T2V methods using only images is insufficient. To comprehensively evaluate the effectiveness of the proposed dynamics metrics, the authors are recommended to include several examples of fake videos generated by existing methods along with their corresponding dynamics scores. This comparison would help in understanding how the new metrics perform relative to current standards and whether they offer a significant improvement in evaluating video quality.

3. **Computational Efficiency**: The computational efficiency of the proposed method is not well addressed. Understanding the resource requirements and processing time for the new dynamics metrics is crucial for practical applications. If the method is computationally intensive and very slow, it might limit its applicability in real-world scenarios where quick and efficient evaluations are necessary.

**Questions:**

Please refer to the weaknesses.

**Limitations:**

This paper discusses the limitations in Section 6.

---

> ### Author Rebuttal · Authors · 2024-08-07
>
> #### **Q1: Demonstrations of Dynamics Score of Real Videos (weakness1) and Fake Videos from Different T2V Methods (weakness2).**
> Thank you for your valuable suggestion. We have collected some real videos and fake videos from different T2V models with dynamic scores for demonstration. However, due to NeurIPS guidelines, we **CAN'T** include links in the rebuttal. Therefore, we present them as images in the PDF instead. Please refer to the attached files. We will also create a project page containing the demo videos in the future for a more intuitive understanding, as you suggested.
>
> As demonstrated in the paper, our dynamics scores correlate with human scores at a level exceeding 90%.
> The improved metrics can better reflect the quality of generation across different levels of dynamics, as measured by root mean square bias.
> As shown in the table below, the improved metrics achieve a **lower root mean square bias** in assessing video quality across various dynamic levels for Motion Smoothness, Subject Consistency, Background Consistency, and Naturalness.
> | Metric | Motion Smoothness | Subject Consistency | Background Consistency | Naturalness |
> |--------|-------------------|---------------------|------------------------|-------------|
> | Improved | 0.44 | 0.41 | 0.43 | 0.29 |
> | Original | 0.56 | 0.55 | 0.56 | 0.39 |
>
>
> #### **Q2: Computational Efficiency**
> The evaluation of proposed dynamics metrics incur low computational cost, processing approximately 10 FPS on a single NVIDIA A100 GPU with support for multiple GPUs. Evaluating models like VideoCrafter 2 takes about 20 minutes, reducible to minutes with parallel GPU processing—a common practice when developing video generation system.

---

> > ### Comment · Reviewer_VYfN · 2024-08-12
> >
> > Thank you for the rebuttal. I appreciate that the authors will also create a project page containing the demo videos in the future for a more intuitive understanding; this will help everyone better grasp the effectiveness of the new evaluation metrics. I would suggest that the authors include the information regarding efficiency in the camera-ready version, as it will aid in understanding the practical utility of the evaluation. I have no further questions. Best of luck to the authors.

---

> ### Comment · Area_Chair_gdzK · 2024-08-12
> **Final thoughts**
>
> Hi Reviewer VYfN
>
> The discussion period is ending soon. Please take a moment to review the author's rebuttal and share any remaining concerns or questions.
>
> Thank you,
> AC

---

> ### Author Response · Authors · 2024-08-13
>
> Thank you for your positive feedback and score. We will ensure to include detailed efficiency information in camera-ready version. Your support is greatly important to us, and we look forward to carrying your good wishes forward. Thanks again.

---

### Official Review · Reviewer_HC61 · 2024-07-12

**Soundness:** 4
**Presentation:** 3
**Contribution:** 3
**Rating:** 7
**Confidence:** 5

**Summary:**

This paper proposes DEVIL, an evaluation suite of metrics for evaluating text-to-video generation, focusing on dynamics (the authors note that many previous works on video generation focus on other aspects but ignore dynamics). Proposed metrics are computed using a number of automatically extracted values based on e.g. optical flow or autocorrelations and then a linear combination of these values are learned such that they align with human perceptions of dynamics.

The authors also rate prompts (automatically using gpt4) to assign a dynamics “grade” which allows them to evaluate alignment with the level of dynamics asked for by a text prompt.  They also report a “dynamic range” of a given model across the set of prompts. Finally the authors use their proposed metrics to evaluate a number of recent video generation models.

**Strengths:**

This paper attacks a useful problem in video generation that researchers talk about / are aware of, but don’t have good metrics to measure.  Overall the proposed metrics seem quite thorough and as shown in the results, the proposed measures do seem to be quite correlated with human judgements.  I think this would be a good contribution to the community particularly if made publicly available.

**Weaknesses:**

Conceptually the novelty of the paper is mostly incremental but since good measures of dynamism in video generation do not currently exist, the work should be of good value to the community.

One potential weakness is that I am not sure if I agree with the decision to segment videos based on length relative to total number of frames since this ignores the fact that different models can generate widely different video lengths or at quite different framerates(e.g. 2 seconds at 8 fps for VideoPoet videos and some examples from Sora that go to 1 min at 24 fps).

There is also sloppy writing in some parts of the paper.  Examples:
* Some tables are never referenced in the body of the text
* Authors mention that human “refine” the dynamics grades but no details are given other than to say that they did this for three months (which is mostly meaningless).
* The word cloud in Figure 3 is not very interpretable — is this for just the high dynamics prompts or everything? What is the take-away?
Eqn 3 (which mentions 5%) does not agree with the paragraph above
* The function SIM is never defined
* Orphaned footnote in line 248
* Plots in Fig 5 are attractive but very hard to actually compare across the different methods

**Questions:**

* I believe that the proposed measures are not necessarily able to separate camera motions from object motions.  Similarly, it is not able to distinguish between fast moving “video textures” (e.g. roaring flame) which tend to be easy to generate using these models vs more specific motions (e.g. ninja doing a backflip).  I would ask the authors to comment on whether these are real limitations or not.
* Suggestion: Please add examples of prompts with high dynamics and low dynamics in the main body of the paper.
* Suggestion: Report linear regression weights (given that there are just a few weights and would allow the reader to evaluate the relative importance of each factor)
* Question: Will the code/prompts be made public?
* Are the “improved metrics” better correlated with human judgements too?  I am not sure if I saw this somewhere in the paper
* Suggestion (but lower priority): I believe Runway has a control that lets you control the dynamics (by changing CFG weight) — it would be interesting to see the effect of this control on measures proposed in this paper.

**Limitations:**

yes

---

> ### Author Rebuttal · Authors · 2024-08-07
>
> #### **Q1: Concern about segmenting videos based on length relative to total number of frames.**
> 1. **Why using relative length.** Segmenting videos by relative length enables standardized comparison no matter how long the video is. The following table shows consistently high correlation values with different ratios, demonstrating our method's robustness.
>
> | ratio | PC   | KC   |
> |-------|------|------|
> | 1/8   | 0.92 | 0.90 |
> | 1/4   | 0.94 | 0.91 |
> | 1/2   | 0.93 | 0.90 |
>
> 2. **Influence of frame rate**: We standardized videos to 8 FPS, following Vbench protocol, to eliminate frame rate variability effects.
>
> 3. **Influence of video length**: We group videos based on the video length (max is 8s in tested models) and study the relation between dynamics scores and human scores. DEVIL robustly achieves over 90% correlation whatever the video length is.
>
> | Video length | PC   | KC   |
> |----------------|------|------|
> | 2s             | 0.96 | 0.94 |
> | 4s             | 0.93 | 0.91 |
> | 8s             | 0.94 | 0.90 |
>
> #### **Q2: Necessity of Distinguishing Motion Types**
> Thanks for highlighting this. Our paper focuses on creating a comprehensive dynamic scoring system for various motions, rather than differentiating specific motion types. In video generation, factors such as object categories, contexts, and actions affect efficacy. Therefore, we designed a benchmark with 19 object categories, 4 scenes, and 10 dynamic categories, covering over 40 subtypes. These subtypes include camera movements, actions, complex effects, and environmental dynamics, ensuring our evaluation system accurately measures video generation performance across diverse scenarios.
>
> #### **Q3: Examples of prompts**
> The following are examples of prompts with different dynamics grades, we will add them to the main paper.
>
> **Static:**
> a man is laying on the ground.
>
> **Low dynamics:**
> A male fencer adjusts his epee mask and prepares to duel with his sparring partner in slow motion.
>
> **Medium dynamics:**
> Tilt up of shirtless sportsman doing pull-ups on bars during cross-training workout at gym.
>
> **High dynamics:**
> A runner explodes out of the starting blocks, racing down the track.
>
> **Very High dynamics:**
> A medieval siege with catapults launching, walls breaking, soldiers charging, and arrows raining down.
>
> #### **Q4: Report linear regression weight**
> The linear regression weight of each dynamics score is as follows, we will add them to the main paper.
>
> | Temporal Scale | Dynamics Score | Typical Value | Weight |
> |----------------|----------------|---------------|--------|
> | Inter-frame | $D_{ofs}$ | 62.00 | 6.70E-04 |
> | | $D_{sd}$ | 1.00 | 0.17 |
> | | $D_{pd}$ | 33.00 | 0.03 |
> | Inter-segment| $D_{pa}$ | 0.80 | 0.63 |
> | | $D_{ga}$ | 0.20 | 2.20 |
> | Video | $D_{te}$ | 7.00E+04 | 1.00E-05 |
> | | $D_{tsd}$ | 0.20 | 1.46 |
>
> #### **Q5: Code release**
> Yes, we will release the code, rompts, and weights.
>
> #### **Q6: Effetiveness of improved metrics.**
> Improved Metrics enhances model performance evaluation through grouping assessment based on video dynamics. Its effectiveness is demonstrated from two perspectives:
> 1. **Representativeness**.
> Improved Metrics better represent a model's ability to generate quality videos with varying dynamics, as shown by lower root mean squared bias compared to Original Metrics. The table below demonstrates this improvement.
>
> | Metric | Motion Smoothness | Subject Consistency | Background Consistency | Naturalness |
> |--------|-------------------|---------------------|------------------------|-------------|
> | Improved | 0.44 | 0.41 | 0.43 | 0.29 |
> | Original | 0.56 | 0.55 | 0.56 | 0.39 |
>
> 2. **Human Correlation**.
> Improved metrics evaluate the model as a whole, not individual videos. To evaluate consistency with human scores, annotators grouped videos by dynamics and assessed their quality(focusing on naturalness due to time limitations). We averaged scores for each group to derive a comprehensive human score per model. The table below compares the human correlation of original and improved metrics.
>
> | Metric | Pearson's Correlation | Kendall's Correlation |
> |--------|----------------------|----------------------|
> | Improved | 0.70 | 0.60 |
> | Origion | -0.60 | -0.33 |
>
> Our improved metric shows a 0.70 Pearson correlation with human scores, confirming its effectiveness. The original metric, dominated by low-dynamic videos, overlooks dynamics and shows a negative correlation with human scores, despite accurate individual video ratings.
>
> #### **Q7: Effetiveness of Runway CFG**
> During Runway GEN-2's evaluation, we considered CFG functionality by assigning different CFG weights based on the prompts' dynamics grade. With such strategy, the model achieves a high degree of dynamics control capability, as evidenced by the high Dynamics Alignment metric.
>
> #### **Q8: Some tables are never refered**
> We will reference the previously uncited Table 1 and Table 2 in Section 3 and Section 4 respectively.
>
> #### **Q9: Details of Human Refinement**
> We have designed detailed criteria and examples for each dynamics grade, used by both human annotators and GPT-4. See Fig. 7 in the Appendix for more information. This will be clarified in the paper.
>
> #### **Q10: Figure 3 is not very interpretable**
>
> 1. Fig. 3 is a word cloud of all prompts in the DEVIL prompt benchmark. It shows that users often use similar terms like "rapid" and "extremely fast" for high-dynamics scenarios, making these words more prominent.
> 2. The number in the preceding paragraph of Eqn 3 should be corrected to 5% to match Equation 3.
>
> #### **Q11: The function SIM is never defined**
> SIM represents the cosine similarity between two feature vectors.
>
> #### **Q12: Orphaned footnote in line 248**
> Thank you for pointing out. We will make the necessary corrections.
>
> #### **Q13: Plots in Fig 5 are attractive but very hard to actually compare across the different methods**
> We'll improve Figure 5 to make it more clear.

---

### Official Review · Reviewer_7oNg · 2024-07-14

**Soundness:** 2
**Presentation:** 3
**Contribution:** 2
**Rating:** 5
**Confidence:** 5

**Summary:**

The paper presents a comprehensive study on the evaluation of Text-to-Video (T2V) generation models, with a particular focus on the dynamics of video content. The authors introduce a novel evaluation protocol named DEVIL, which aims to address the often overlooked aspect of dynamics in existing evaluation metrics. This protocol defines a set of dynamics scores that correspond to multiple temporal granularities and introduces a new benchmark of text prompts under various dynamics grades.

In addition, this paper verifies a significant issue with some existing evaluation metrics: they allow models to attain high scores by generating low-dynamic videos. The authors demonstrate that the metrics provided by DEVIL align closely with human in rating dynamics, indicating its potential to significantly advance T2V generation models. The paper also highlights the current limitations in T2V models and datasets, and provides valuable suggestions for future research in this field.

**Strengths:**

1. The paper introduces an innovative T2V model metric, DEVIL, which evaluates the content dynamics of generated videos—an aspect often overlooked by current evaluation methods but essential for realistic video generation. DEVIL assesses dynamics at multiple granularities, achieving high correlation with human evaluations. By revealing the negative correlation between existing metrics and dynamic scores, the paper challenges the current evaluation standards.
2. The study employs robust experiments to validate the dynamics metric (DEVIL), achieving up to 90% consistency with human evaluations. Additionally, the paper identifies issues with existing T2V evaluation metrics, revealing their negative correlation with dynamics metrics and the tendency to favor low-dynamic content, which misrepresents model performance.
3. The theoretical framework is detailed and thorough, covering the formulation of dynamics scores across multiple temporal granularities. The paper also highlights a reliable human rating collection process, amassing 50,000 prompt dynamic level classifications for training the linear regression model used in the metrics and evaluations of 4,800 generated videos from six T2V models for proving the consistency between the metric and human ratings.
4. The paper is overall logically structured and easy to follow.

**Weaknesses:**

1. How to differentiate dynamic video from videos with low-quality motions, for example, flickering or temporal inconsistency? We desire videos with not only large dynamics but also high-quality dynamics, but some models which generate low-quality motion such as flickring frames and temporally-inconsistent videos may also achieve high dynamic scores under the proposed evaluation scheme.

2. Some previous works also propose metrics to evaluate dynamic levels of generated videos. For example, VBench [23] proposes dynamic degree evaluated with RAFT, and EvalCrafter [31] proposes motion quality evaluated with RAFT. The proposed metric should be compared with existing metrics from those works.

3. Frame rate of videos is an important factor that may affect the metrics. For example, a video with low fps may get higher score than a model with high fps if the evaluation is conducted on all frames without aligning their fps. Do the authors take it into consideration when designing the metrics? Are the models evaluated at their original fps or samped frames at the same fps?

4. In Section 3.3, how many videos are used to fit the human alignment dynamic scores? I am concerned that the metric is overfitting the selected videos for human alignment, but may not generalize well to other videos. Are the human evaluation results in Table 3 computed from the same videos as the videos used to fit the dynamic scores in Section 3.3? There is a risk of cheating if using the same set of videos to fit the human alignment dynamic score (in Section 3.3) and to calculate the human correlation (in Table 3).
5. The effectiveness of the improved metrics proposed in Section 4 is not validated with human correlation.
6. The relationship between the naturalness score defined in Section 4 and the rest of the paper is not clear.

**Questions:**

Please refer to weakness section.

**Limitations:**

The authors addressed the limitations and societal impact in the paper.

---

> ### Author Rebuttal · Authors · 2024-08-07
>
> #### **Q1: How to differentiate dynamic video from videos with low-quality motions?**
> We aim to motivate models to produce videos with a wide range of dynamics while maintaining high quality. We recognize that some videos may exhibit high dynamics but poor quality.
> To address, we have enhanced quality metrics by integrating dynamics scores, enabling a more comprehensive evaluation of models' generative capabilities rather than assessing quality and dynamics independently.
> When filtering individual videos based on both quality and dynamics, our dynamics scores are complementary to exisiting video quality scores.
>
> #### **Q2: Compare with dynamics degree in Vbench and motion quality in EvalCrafter**
>  Thank you for this insightful comment. DEVIL is superior to existing T2V evaluation protocols in three key aspects.
>
> 1. Dynamics Assessment
> - Vbench and EvalCrafter are bulit solely on optical flow, limiting analysis to inter-frame pixel motion dynamics. They fail to capture non-pixelmotion dynamics, such as variations in illumination, style, and color.
> - DEVIL employs seven dynamics scores across three temporal scales (inter-frame, inter-segment, and video level) to assess video dynamics comprehensively. It significantly improves correlation with human evaluation, achieving an 84% win ratio for inter-frame dynamics, substantially outperforming optical flow's 73%.
> 2. Evaluation of Models' Capabilities on Dynamics
> - Vbench and EvalCrafter created Dynamic Degree and Flow Score metrics based on optical flow, both favoring videos with greater dynamics. However, they neglect the importance of matching video dynamics to prompt requirements, which can range from low to high.
> - EvalCrafter's Motion AC-Score assesses a model's ability to generate videos with low/high dynamics according to prompts. However, as a binary metric, it only roughly measures dynamic controllability and doesn't capture the full range of a model's dynamic capabilities.
> - We propose the Dynamics Range, $M_{range}$, to drive generating videos with various dynamics, and Dynamics Alignment, $M_{align}$ to evaluate how well models can control dynamics based on text prompts. Our approach enables a more reasonable and precise assessment of a model's dynamic capabilities.
> 3. Quality Metrics
> - Vbench and EvalCrafter assess video dynamics and quality independently, neglecting their correlation. As models tend to generate low-dynamics videos, quality metrics are skewed towards these, failing to accurately represent model performance across the full range of dynamics.
> - EvalCrafter's Action Recognition assesses human actions using recognition scores, but it's limited to human activities and fails to evaluate model quality across various dynamics ranges.
> - We combine dynamics with existing quality metrics to quantitatively assess models' ability to generate high-quality videos across dynamic ranges, enabling more comprehensive evaluation.
>
> #### **Q3: Influence of Frame Rate**
> Thanks for the constructive comment. In the paper, we DO standardize the frame rate of each video to 8 FPS, following Vbench. Experiments show our dynamics evaluation maintains high correlation (>0.9) with human ratings across various frame rates. The table below shows Pearson correlation between our dynamics scores and human ratings:
>
> | Dynamics    | 4FPS  | 8FPS  | 16FPS | Origin FPS |
> |-------------|-------|-------|-------|------------|
> | Inter-frame | 0.952 | 0.950 | 0.946 | 0.951      |
> | Inter-Segm  | 0.952 | 0.954 | 0.954 | 0.953      |
> | Video       | 0.967 | 0.967 | 0.967 | 0.967      |
>
> #### **Q4: Details of human alignment**
> We used a 75/25 split of the annotated data, 75%(3600 videos) for fitting the human alignment dynamic scores, 25%(1200 videos) reserved as a test set. Results in Table 3 are computed exclusively from the 25% test set.
>
> #### **Q5: Effectivenss of the improved metrics**
> Improved Metrics aims to achieve a more comprehensive and accurate evaluation of model performance by employing grouping assessment based on video dynamics. The effectiveness of Improved Metrics can be demonstrated from two perspectives:
> 1. Representativeness
>
> Improved Metrics can better represent models' capability to generate high-quality videos with varying levels of dynamics, as measured by the root mean squared bias. As the table below demonstrates, our Improved Metrics achieve a **lower root mean squared bias** than the Original Metrics.
>
> | Metric | Motion Smoothness | Subject Consistency | Background Consistency | Naturalness |
> |--------|-------------------|---------------------|------------------------|-------------|
> | Improved | 0.44 | 0.41 | 0.43 | 0.29 |
> | Original | 0.56 | 0.55 | 0.56 | 0.39 |
>
> 2. Human Correlation
>
> Our metrics are defined for the entire model rather than individual videos. Annotators grouped videos by their dynamics and assessed their quality, focusing on naturalness due to time limitations. We then averaged the scores for each dynamic group to obtain a comprehensive human score for each model. The table below shows the human correlation evaluation of the original metrics and the improved metrics.
>
> | Metric | Pearson's Correlation | Kendall's Correlation |
> |--------|----------------------|----------------------|
> | Improved | 0.70 | 0.60 |
> | Original | -0.60 | -0.33 |
>
> Our improved metric achieved a Pearson correlation of 0.70 with human scores, confirming its effectiveness. The original metric, which averages video quality without considering dynamics, is dominated by low-dynamic videos. This oversight leads to a negative correlation with human scores, despite accurate individual video ratings.
>
> #### **Q6: Naturalness**
> Naturalness is inspired by the observation that dynamics also accompany unnatural scenarios, e.g., car wheels spinning rapidly while the vehicle remains stationary. We thus propose naturalness to reflect how much the generated videos are like camera-captured ones.  Note that it achieved a correlation of 79% with human ratings.

---

> > ### Comment · Reviewer_7oNg · 2024-08-13
> > **Official Comment by Reviewer 7oNg**
> >
> > Thank the authors for the rebuttal. The authors addressed most of my concerns so I raised my rating to borderline accept.

---

> > > ### Author Response · Authors · 2024-08-13
> > >
> > > Thank you so much for your positive rating!

---

> ### Comment · Area_Chair_gdzK · 2024-08-12
> **Final thoughts**
>
> Hi Reviewer 7oNg,
>
> The discussion period is ending soon. Please take a moment to review the author's rebuttal and share any remaining concerns or questions.
>
> Thank you,
> AC

---

> ### Author Response · Authors · 2024-08-13
>
> Before the end of the rebuttal, we look forward to your valuable feedback and further suggestions. We will take full efforts to address your concerns. Looking forward to your reply. Thank you very much.

---

### Author Rebuttal · Authors · 2024-08-07

Thanks to all reviewers and ACs for the valuable comments and suggestions. In the original review, all the reviewers acknowledged the contributions of the proposed evaluation protocol(DEVIL). The strengths are summarized as follows:

1. Reviewer R-7oNg: "The paper introduces an **innovative** T2V model metric", "the theoretical framework is detailed and thorough", "employs **robust experiments**... achieving up to **90% consistency** with human evaluations", "The paper is overall logically structured and easy to follow."
2. Reviewer  R-HC61: "This paper attacks a **useful problem** in video generation that **researchers talk about / are aware of**, but don’t have good metrics to measure", "quite **correlated with human judgements**", "**good contribution** to the community".
3. Reviewer R-VYfN: "The study presents a **novel** set of dynamics metrics", "offers a **more detailed assessment** of T2V models".
4. Reviewer  R-Kt6U: "It explores a **more fine-grained protocol**", "propose the 'improved metric'... would be **useful** "

The major concerns and suggestions are mostly about novelty (R-7oNg, R-Kr6U), effectiveness of improved metrics (R-7oNg, R-HC61), computation efficiency (R-VYfN), technical details (R-7oNg, R-HC61) and paper writing (R-HC61, R-Kt6U).

During the rebuttal, we carefully considered the reviewers' feedback and provided results of more experiments. We believe our responses can address reviewers' concerns and enhance our work.

---

### Decision · Program_Chairs · 2024-09-25

**Decision:**

Accept (poster)

**Comment:**

This paper examines the evaluation protocols for text-to-video generation. Initially, the reviewers identified weaknesses in assessing motion dynamics, the fairness of comparing FPS, comparisons with existing metrics for evaluating dynamics, and the details about human evaluation. In the rebuttal, the authors clarified several points and provided additional results to support their claims, leading two reviewers to raise their ratings to borderline accept. Ultimately, no reviewers opposed the acceptance of this work. The remaining expectation is for the authors to incorporate their changes in the rebuttal into their camera-ready version. Therefore, the AC recommends accepting this submission.